

# Unsupervised anomaly detection of permanent magnet offshore wind generators through electrical and electromagnetic measurements

Ali Dibaj[1], Mostafa Valavi[2], and Amir R. Nejad[1]

[1]Department of Marine Technology, Norwegian University of Science and Technology (NTNU), Trondheim, Norway
[2]EDR & Medeso AS, Oslo, Norway

**Correspondence:** Ali Dibaj (ali.dibaj@ntnu.no)

**Abstract.** This paper investigates fault detection in offshore wind permanent magnet synchronous generators (PMSG) for demagnetization and eccentricity faults (both static and dynamic) at various severity levels. The study utilizes a high-speed PMSG model, on the NREL 5-MW reference offshore wind turbine, and at the rated wind speed, to simulate healthy and faulty conditions. An unsupervised convolutional autoencoder (CAE) model, trained on simulated signals from the generator in its

healthy state, serves for anomaly detection. The main aim of the paper is to evaluate the possibility of fault detection by means of high-resolution electrical and electromagnetic signals, given that the typically low-resolution standard measurements used in SCADA systems of wind turbines often impede the early detection of incipient failures. Signals analyzed include three-phase currents, induced shaft voltage, electromagnetic torque, and magnetic flux (airgap and stray) from different directions and positions. The performance of CAE models is compared across time and frequency domains. Results show that in the time

domain, stator three-phase currents effectively detect faults. In the frequency domain, stray flux measurements, positioned at the top, bottom, and sides of outside the stator housing, demonstrate superior performance in fault detection and sensitivity to fault severity levels. Particularly, radial components of stray flux can successfully distinguish between eccentricity and demagnetization.

## 1   Introduction

Permanent magnet synchronous generators (PMSGs) have been recently popular in offshore wind applications driven by advancements in permanent magnet materials and high-efficiency power electronics. Figure 1, sourced from the Global Offshore Wind Report 2022 (GWEC, 2022), illustrates the evolution of drivetrain technologies in offshore wind turbines within the European and Chinese markets from 2016 to 2021. The data indicate that in 2016, the market share of PMSGs was approximately 60% in Europe and 10% in China. By 2021, these shares had risen significantly, with PMSGs accounting for 100% of the

market share in Europe and 80% in China.

Distinct from traditional doubly-fed induction generators (DFIGs), PMSGs leverage permanent magnets to generate the magnetic field, thereby eliminating the need for a separate excitation system. This design eliminates components like slip rings and brushes, leading to higher reliability and reducing maintenance requirements. Available in both medium-speed and direct-

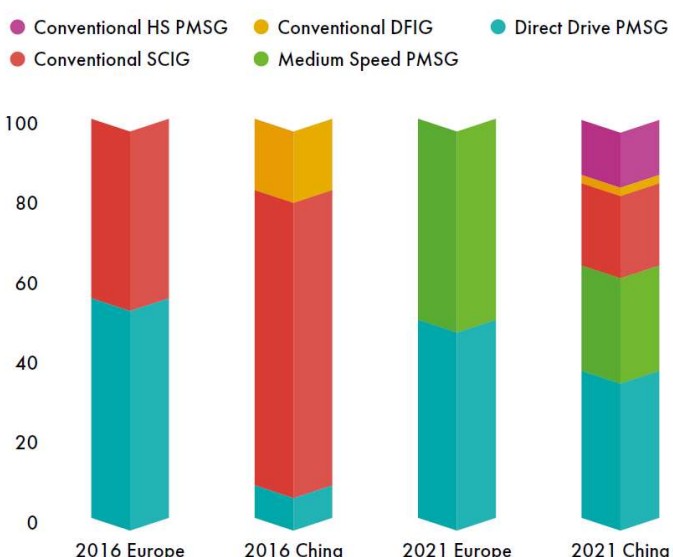

**Figure 1.** Trend of offshore wind turbine drivetrain technology between 2016 and 2021 (GWEC, 2022)

drive configurations, these generators offer higher power density and efficiency, with added benefits such as improved grid
stability because of a faster response to wind speed changes (Moghadam and Nejad, 2020; Freire and Cardoso, 2021; Carroll
et al., 2015). However, despite their increased reliability, PMSGs are not completely immune to faults and have their own
challenges, particularly in harsh offshore environments. Consequently, reliable condition monitoring and early fault detection
are essential to minimize production loss and prevent unexpected downtime in these machines (Nejad et al., 2022; Huang et al.,
2023; Yang, 2009; Mahmoud et al., 2024).

PM machines, in general, are susceptible to several types of faults that can impact their functionality (Choi et al., 2018;
Kudelina et al., 2021). Stator failures are common, which can include insulation faults and issues with the connections in
the stator windings (Wang et al., 2014; Nyanteh et al., 2013; Hoang Nguyen et al., 2023; Ortiz-Medina et al., 2023). Airgap
eccentricity can cause operational disturbances such as vibrations and noise and may lead to mechanical stress and uneven
wear (Valavi et al., 2013; Ebrahimi et al., 2009, 2014; Tong et al., 2020). Demagnetization of the permanent magnets is also a
critical fault, often triggered by excessive heat, leading to a permanent reduction in the generator's efficiency and power output
(Faiz and Mazaheri-Tehrani, 2017; Huang et al., 2023; Ebrahimi et al., 2022; Wang et al., 2016). Additionally, failures in the
cooling and control systems can significantly impact the PMSG's performance (Borchersen and Kinnaert, 2016). The cooling
system is crucial for maintaining an optimal operational temperature and preventing overheating, while the control system
manages the generator's operational parameters (Freire and Cardoso, 2021).

This study examines the problems of demagnetization and eccentricity failures, which are commonly encountered in PMSGs.
Currently, there is a lack of established techniques to effectively address these issues and meet the requirements of the wind





sector. The demagnetization issue is exacerbated in offshore environments where thermal management is more difficult due to excessive humidity, maintenance, and accessibility concerns (Gyftakis et al., 2023). Demagnetization in PMSGs can be either local or distributed, each showing distinct fault signatures. Local demagnetization refers to the loss of magnetic properties in

specific areas of the magnetic poles, often due to localized overheating or physical damage. Distributed demagnetization, on the other hand, involves a uniform reduction in magnetic strength across the entire magnet, typically resulting from prolonged exposure to high temperatures or electrical faults (Choi et al., 2018; Choi, 2021). Eccentricity is also characterized as static and dynamic. Static eccentricity is characterized by a constant offset between the rotor and stator, leading to an uneven magnetic field and potentially causing vibrations and wear. Dynamic eccentricity involves a varying distance between the rotor and stator

during rotation, which can result in fluctuating magnetic forces, additional stress on bearings, and operational instability (Freire and Cardoso, 2021; Kudelina et al., 2021).

Various monitoring techniques are employed for the purpose of condition monitoring and fault detection of electrical machines, including PMSGs, depending on the specific type of failure. Vibration analysis, which commonly utilizes high-resolution accelerometer data (Dibaj et al., 2022, 2023), is performed to identify defects such as mechanical unbalance and

55 bearing damage (Ágoston, 2015; Ali et al., 2019; Ding et al., 2022), eccentricity (Ogidi et al., 2015; Su and Chong, 2007), and electrical faults (Singh and Sa'ad Ahmed, 2004; Su et al., 2011). However, early-stage electrical and electromagnetic faults do not often produce significant mechanical vibrations and, therefore, are not easily detectable from vibration signatures. Temperature monitoring techniques such as Supervisory Control and Data Acquisition (SCADA) system (Zhao et al., 2017; Qiu et al., 2016) or infrared thermography (Stipetic et al., 2012; Lopez-Perez and Antonino-Daviu, 2017) can detect problems related to

60 bearings (Choudhary et al., 2021), short circuits in stator coils (Khanjani and Ezoji, 2021), and cooling systems (Borchersen and Kinnaert, 2016). However, temperature-based methods face challenges, including difficulty in sensor placement to accurately identify specific faults, the sensor's general sensitivity that might only offer a broad temperature overview rather than detailed hotspots, and the potential influence of environmental conditions on temperature readings, which may affect the precise identification of problems. Furthermore, temperature measurements, as a part of SCADA systems, are unable to capture

fast dynamics and provide fault discrimination to the required level because of low-resolution data.

In this work, electrical and electromagnetic signals, including stator phase currents, induced shaft voltage, electromagnetic torque, and magnetic flux density inside and outside the airgap, are analyzed and compared for fault detection in PMSG. Harmonic analysis of electrical and electromagnetic signals is a common technique for identifying faults in PM machines across various industries and applications, as supported by various studies (Valavi et al., 2018, 2013; Bernier et al., 2023; Da

et al., 2013; Zhang et al., 2021). Furthermore, advanced signal processing methods such as Wavelet Transform (Ehya et al., 2022) and Hilbert Huang Transform (Zhang et al., 2021) are also used to extract harmonic characteristics from these signals. Despite these approaches, the effectiveness of these measurements, particularly the electromagnetic ones, in providing early failure warnings in large MW-scale offshore PMSGs has yet to be established. Additionally, it is important to note that certain measurements examined in this work, such as phase currents, might also be integrated into the SCADA systems of wind

turbines. However, the raw data of these measurements are typically downsampled in SCADA systems to a 10-min resolution.



The main drawback with the downsampled data of SCADA systems is that they cannot pinpoint incipient failures in PMSG as early as possible, highlighting an essential area for research on the capabilities of these measurements.

Recent advancements in computational power and cloud computing have significantly shifted industrial asset management towards machine learning and artificial intelligence techniques (Peres et al., 2020; Lei et al., 2020). This shift aims to address the inefficiencies of traditional data management methods in handling the vast amounts of data involved in large-scale applications like offshore wind farms. Moreover, machine learning models are known for their scalability and flexibility (Lu et al., 2024). They don't have the limitation of traditional methods in maintaining accuracy as the scale of data and model complexity increases. Despite these advantages, the application of machine learning for fault detection in PMSGs is still largely unexplored (Freire and Cardoso, 2021).

Therefore, this study adopts a machine learning model for unsupervised anomaly detection, trained on collected simulated measurements in the healthy state. Unlike supervised learning, which requires a significant amount of labeled data with predefined class labels for training, unsupervised learning does not rely on labeled data. This characteristic is particularly beneficial in offshore wind applications, where acquiring extensive labeled fault-related data is challenging. Moreover, supervised learning methods often struggle to generalize to unseen fault scenarios. Consequently, a convolutional autoencoder (CAE) model is utilized, known for its capability to process complex and high-dimensional data efficiently.

In summary, this study aims to conduct a comparative analysis of different measurements—three-phase currents, induced shaft voltage, electromagnetic torque, and airgap and stray magnetic flux density—for the purpose of wind turbine PMSG anomaly detection using a CAE model. As mentioned earlier, despite the potential availability of a few of these measurements in SCADA systems, they are often recorded at a low resolution, typically every 10 minutes. The primary focus of this work is to explore the effectiveness of high-resolution measurements for the early detection of potential failures. The sensitivity of these measurements against the studied fault cases, including demagnetization and static and dynamic eccentricity at various fault severity, will be analyzed. Simulated measurements will be collected from a simulation high-speed PMSG, designed and modeled based on the specifications of the NREL 5-MW reference offshore wind turbine (Jonkman et al.), tailored for offshore wind applications.

The remainder of the paper is structured as follows: Section 2 describes the generator model, measurements, and studied fault cases. Section 3 discusses the anomaly detection methodology employed in this study, including the CAE model, threshold determination, and performance metrics. Section 4 contains the results and discussion. Finally, the conclusion is outlined in Section 5.

## 2 Generator model

A wind generator was designed according to the specifications and requirements detailed in "Definition of a 5-MW Reference Wind Turbine for Offshore System Development," a technical report published by the National Renewable Energy Laboratory (NREL) (Jonkman et al.). Table 1 presents the drivetrain specifications as outlined in the report.





**Table 1.** Drivetrain specification (Jonkman et al.)

| Parameter | Value |
| --- | --- |
| Rated rotor speed (rpm) | 12.1 |
| Rated generator speed (rpm) | 1173.7 |
| Gearbox ratio | 1:97 |
| Electrical generator efficiency | 94.4% |
| Generator inertia about high-speed shaft (kg.m$^2$) | 534.116 |
| Equivalent drive-shaft torsional-spring constant (kN.m/rad) | 867637 |
| Equivalent drive-shaft torsional-damping constant (kN.m/(rad/s)) | 6215 |
| Fully-deployed high-speed shaft brake torque (N.m) | 28116.2 |
| High-speed shaft brake time constant (sec) | 0.6 |

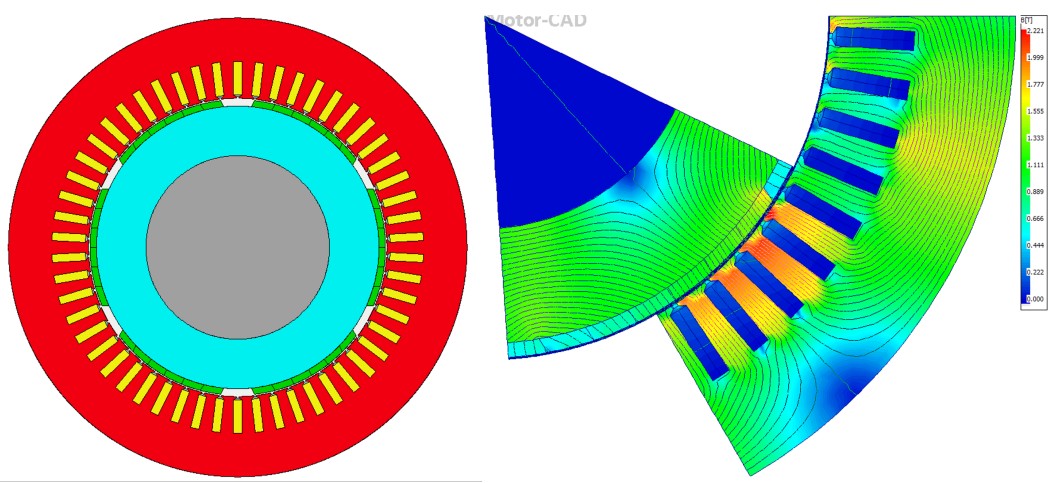

**Figure 2.** Radial Cross-Section and Magnetic Field Distribution of the Generator

Following these specifications, a high-speed PMSG was developed and optimized. Figure 2 illustrates the two-dimensional (2-D) cross-section of the designed PMSG along with its magnetic field distribution. The analysis of the generator's performance was conducted using ANSYS Motor-CAD, a specialized electrical machine design software. The generator features a surface-mounted permanent magnet configuration with six poles and 54 slots. To minimize eddy current losses, the permanent magnet blocks are segmented both radially and axially, which enhances efficiency and reduces the risk of demagnetization.

The wind generator is capable of producing 5.177 MW of electromagnetic power at the rated speed, achieving an efficiency of 98.74%. The line current and voltage are measured at 1092 Arms and 3184 Vrms, respectively. A performance analysis confirms that the generator meets the target power output with exceptional efficiency.



**Table 2.** Simulated electrical and electromagnetic measurements

|  | Measurement |
|---|---|
| $Vsh$ | Induced shaft voltage |
| $Te$ | Electromagnetic torque |
| $Is$ | Stator three phase currents (3 signals) |
| $SFr5$ | Stray flux sensor - radial component, outside stator housing with a distance of 5mm (top, bottom, and side locations - 4 signals) |
| $SFt5$ | Stray flux sensor - tangential component, outside stator housing with a distance of 5mm (top, bottom, and side locations - 4 signals) |
| $SFr10$ | Stray flux sensor - radial component, outside stator housing with a distance of 10mm (top, bottom, and side locations - 4 signals) |
| $SFt10$ | Stray flux sensor - tangential component, outside stator housing with a distance of 10mm (top, bottom, and side locations - 4 signals) |
| $AFrt$ | Airgap flux sensor - radial component at tooth position (top, bottom, and side locations - 4 signals) |
| $AFtt$ | Airgap flux sensor - tangential component at tooth position (top, bottom, and side locations - 4 signals) |
| $AFrs$ | Airgap flux sensor - radial component at slot position (top, bottom, and side locations - 4 signals) |
| $AFts$ | Airgap flux sensor - tangential component at slot position (top, bottom, and side locations - 4 signals) |

## 2.1 Measurements

In this study, a collection of simulated measurements from the PMSG model, as detailed in Table 2, are used in the anomaly detection task aimed at identifying fault cases outlined in Section 2.2. This set of collected data includes high-resolution induced shaft voltage, electromagnetic torque, stator phase currents, as well as airgap and stray magnetic flux, including both radial and tangential components at various positions. Variations observed in these measurements can indicate different types of faults depending on the symptoms manifested in the signal. For instance, flux sensors provide insights into the behavior of the magnetic field. Faults that introduce imbalances or irregularities to the rotating magnetic field can be identified using these sensors. Flux monitoring has been recently popular thanks to advancements in sensor technology and low-cost and compact flux sensors such as search coils and hall-effect sensors (Mazaheri-Tehrani and Faiz, 2022). Figure 3 indicates the position of flux sensors implemented in the simulation model in this study. Also, Figure 4 shows some examples of the time-domain waveform of simulated signals. As explained earlier, the motivation of this study extends to comparing the performance of the anomaly detection model trained with these simulated high-resolution signals to examine the diagnostic capabilities of each measurement type.

## 2.2 Fault cases

Two main categories of faults in the PMSG model are considered: partial demagnetization and eccentricity, which includes both static and dynamic forms. The investigation is on a series of fault cases within these categories. The aim is to evaluate the performance of anomaly detection method in varying degrees of fault severity using different measurement variables.

Partial demagnetization faults, outlined in Table 3, are explored through six distinct scenarios, each simulating varying levels of magnetic flux density reduction across the generator's permanent magnets. The first and second cases, FC1 and FC2, model



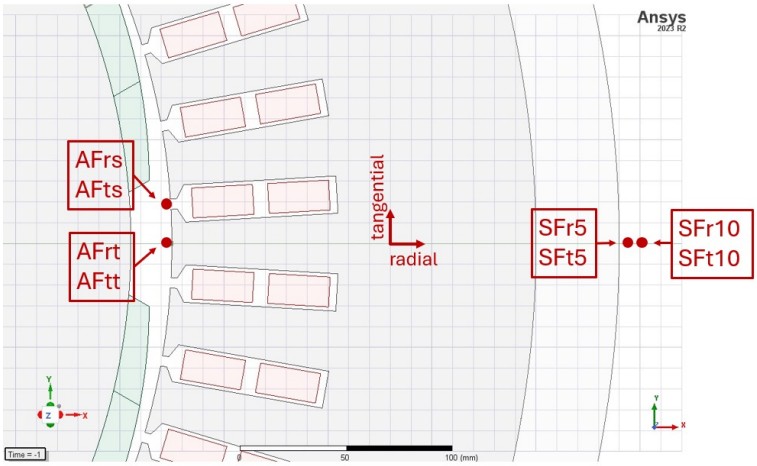

**Figure 3.** Position of the flux sensors in the simulation wind PM generator model

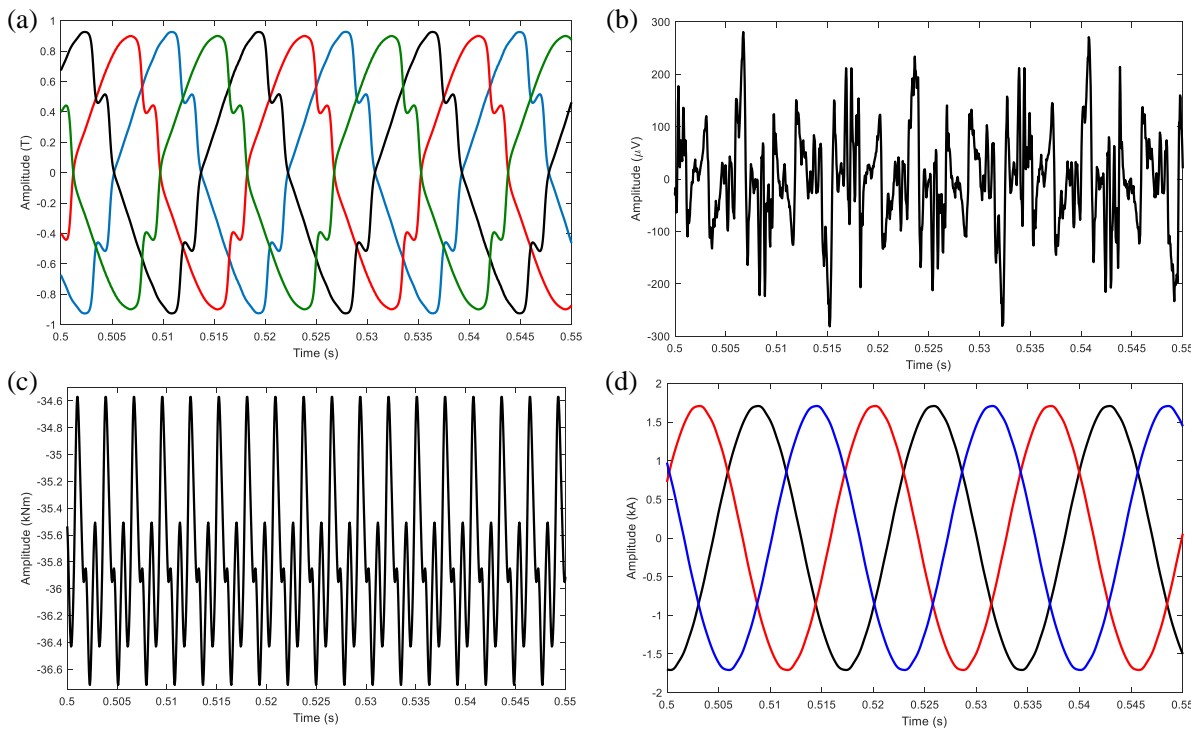

**Figure 4.** Time-domain waveform of the simulated data: (a) radial components of airgap flux (top, bottom, and side locations), (b) induced shaft voltage, (c) electromagnetic torque, (d) stator three-phase currents



**Table 3.** Partial demagnetization with six cases

|  | Fault description |
| --- | --- |
| Fault case 1 (FC1) | One pole all segments (10%) |
| Fault case 2 (FC2) | One pole all segments (20%) |
| Fault case 3 (FC3) | 2 segments of one pole (40% & 20%) |
| Fault case 4 (FC4) | 2 segments of one pole (80% & 40%) |
| Fault case 5 (FC5) | 2 segments of all poles (20% & 10%) |
| Fault case 6 (FC6) | 2 segments of all poles (40% & 20%) |

**Table 4.** Static (SE) and dynamic (DE) eccentricity with six different degrees

|  | Fault description |
| --- | --- |
| Fault case 1 (FC1) | Dynamic (5%) |
| Fault case 2 (FC2) | Dynamic (15%) |
| Fault case 3 (FC3) | Dynamic (25%) |
| Fault case 4 (FC4) | Static (5%) |
| Fault case 5 (FC5) | Static (15%) |
| Fault case 6 (FC6) | Static (25%) |

a mild uniform 10% and 20% demagnetization affecting all segments of a single pole. FC3 and FC4 model more localized demagnetization, where only two segments of a single pole are demagnetized at different severities, 40%, and 20% for FC3 and 80% and 40% for FC4, respectively. FC5 and FC6 extend this localized demagnetization to multiple poles, with two segments of all poles undergoing demagnetization at 20% and 10% for FC5 and 40% and 20% for FC6, respectively.

     Eccentricity, characterized by the misalignment of the rotor relative to the stator, is assessed through static and dynamic con-
140 ditions across six cases as shown in Table 4. Dynamic eccentricity, from FC1 to FC3, addresses a variable misalignment where the rotor's axis orbits around the stator's axis at severities of 5%, 15%, and 25%, respectively. Conversely, static eccentricity cases, FC4 to FC6, examine the impact of a fixed rotor offset from the stator axis, also at severities of 5%, 15%, and 25%, respectively.

## 3   Methodology

### 3.1   Convolutional Autoencoders

Autoencoders (AE), developed originally as neural network models for copying input to output, have significantly evolved to play a crucial role in unsupervised learning, dimensionality reduction, and data denoising (Goodfellow et al., 2016). In anomaly


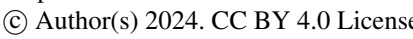

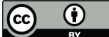

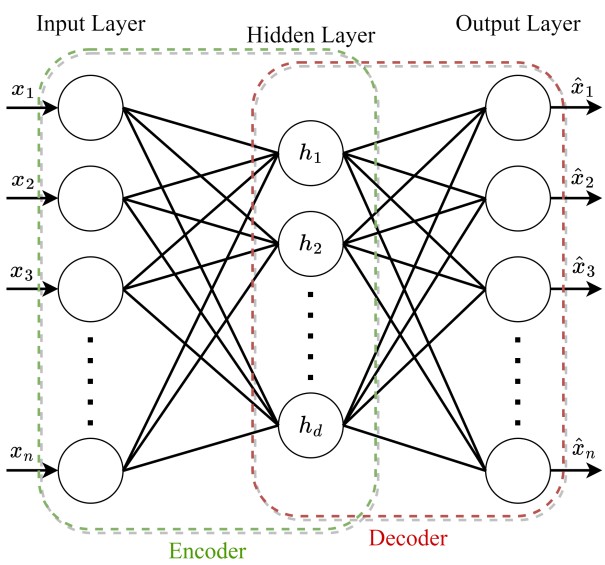

**Figure 5.** Typical autoencoder architecture

detection, AEs are particularly effective; they are trained on normal data to learn its representation, and anomalies are identified based on the higher reconstruction error when the model encounters data that deviates from this learned normal behavior. This higher reconstruction error is because the AE, trained on normal or healthy data, finds it challenging to reconstruct these new or deviant patterns. As mentioned previously, the unsupervised approach is advantageous in anomaly detection in wind turbine applications where anomalies are rare and often not labeled.

A typical AE consists of two main parts: the encoder and the decoder. The encoder compresses the input data as a sequence of data points $\mathbf{x} = [x_1, x_2, x_3, ..., x_n]$ into a lower-dimensional representation known as feature space $\mathbf{h} = [h_1, h_2, h_3, ..., h_d] (d < n)$, and the decoder reconstructs the data back to its original form $\hat{\mathbf{x}} = [\hat{x}_1, \hat{x}_2, \hat{x}_3, ..., \hat{x}_n]$ from this compressed representation as shown in Fig. 5. The formulation for a single-layer encoder and a single-layer decoder is described as follows:

$$\mathbf{h} = f(\mathbf{x}) = s(\mathbf{W}^1 \mathbf{x} + \mathbf{b}^1) \tag{1}$$

$$\hat{\mathbf{x}} = f(\mathbf{h}) = s(\mathbf{W}^2 \mathbf{h} + \mathbf{b}^2) \tag{2}$$

where $\mathbf{W}^1$ and $\mathbf{W}^2$ are weight matrices, $\mathbf{b}^1$ and $\mathbf{b}^2$ are bias vectors. $s(.)$ is the activation function, which is commonly a Sigmoid function $\sigma(t) = 1/(1 + e^{-t})$ or a Rectified Linear Unit (ReLU) function $\text{ReLU}(t) = \max(0, t)$ for the encoder and decoder parts. For the output layer, this function can be a Sigmoid or a linear function, depending on the type of input data (Wu et al., 2021). The architecture of AE is adaptable, allowing for the modification of the number and size of its hidden layers to suit the complexity of the input data (Li et al., 2021). Given a set of training data $\{\mathbf{x}^{(i)}\}_{i=1}^{N}$, the AE model is typically trained





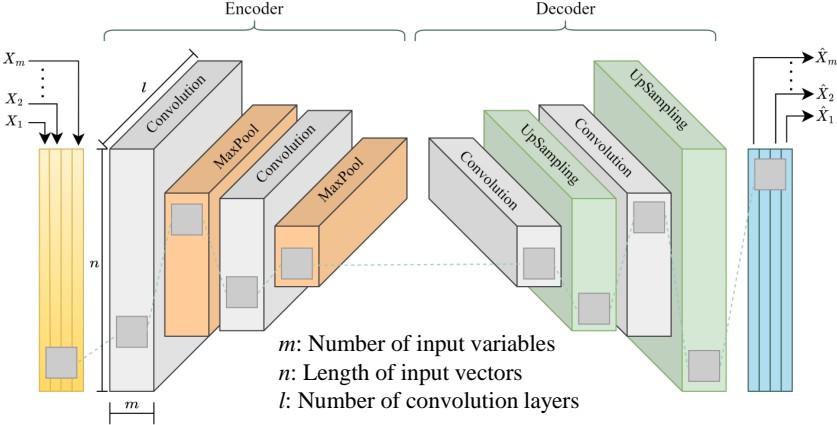

**Figure 6.** Convolutional autoencoder architecture

by minimizing the cost function $J$, often measured by mean squared error (MSE), through the back-propagation algorithm
(Rumelhart et al., 1986), expressed as:

$$J_{MSE}(\mathbf{W}, \mathbf{b}) = \frac{1}{N} \sum_{i=1}^{N} \|\mathbf{x}_i - \hat{\mathbf{x}}_i\|^2 \tag{3}$$

This study employs multi-variable measurements like magnetic flux density measured at different angles. For such data,
a standard one-dimensional AE will not work. In addition, the typical feed-forward AE doesn't take into account the spatial
structure of data, therefore reducing the accuracy of the reconstruction process. To solve this, the convolutional autoencoder
(CAE) model is used in this study. The CAE uses convolutional and deconvolutional layers instead of the fully connected layers
found in the regular feed-forward autoencoder, as shown in Figure 6.

The first part of the CAE works by compressing the data using a series of steps that involve convolution and pooling.
The convolution layers perform operations that apply a filter to the input, which helps to capture important parts of the data
depending on the filter used. After each convolution layer, a pooling step follows. This step usually uses a max-pooling layer,
which reduces the size of the output from the convolution by picking the highest value from each segment of the input data
covered by the filter. The second part of the CAE is about decoding the features that the first part extracted. This is done with
deconvolutional layers. These layers increase the size of the input through a special convolution process to rebuild the input
data in the output, making sure it's the same size as it was originally.

Figures 7 (a)-(d) provide examples of this reconstruction process using induced shaft voltage measurements in both time-
180 domain and frequency-domain representations. Notably, as illustrated in Figures 7 (b) and (d), the CAE model struggles to
reconstruct instances of faulty data accurately. This discrepancy results in a noticeable error between the original and recon-
structed data, highlighting this approach's capability to identify anomalies.



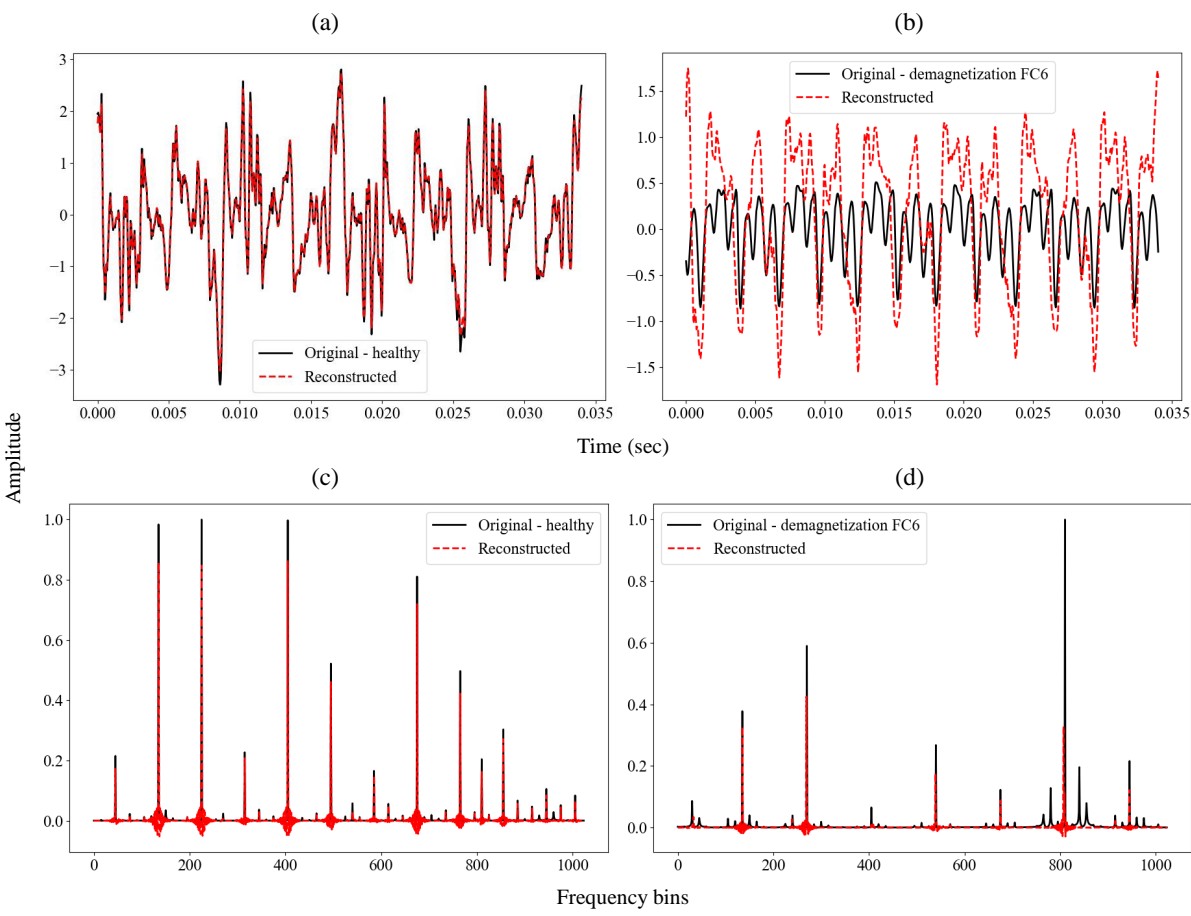

**Figure 7.** Comparison between original input and reconstructed output of CAE model for induced shaft voltage signal: (a) time-domain/healthy, (b) time-domain/demagnetization FC6, (c) frequency-domain/healthy, and (d) frequency-domain/demagnetization FC6.

## 3.2 Threshold determination

For anomaly detection using the reconstruction error obtained by the CAE model, there should be a fault threshold to differentiate between healthy and faulty (anomalies) cases. In this study, the fault threshold is established based on the maximum reconstruction error observed in the training dataset. Existing works on data-driven anomaly detection across various applications have established similar fault thresholds based on the reconstruction error for healthy training data (Chen et al., 2021; Xiang et al., 2022; Campoverde-Vilela et al., 2023; Givnan et al., 2022). The training data comprises instances representing the healthy state of the PMSG. As mentioned, each training sample $\mathbf{x}^{(i)}$ is passed through the CAE model to obtain a reconstructed output $\hat{\mathbf{x}}^{(i)}$. The discrepancy between the original and reconstructed data points, quantified using the MSE cost function, serves as the reconstruction error $e^{(i)} = \text{MSE}(\mathbf{x}^{(i)}, \hat{\mathbf{x}}^{(i)})$. The fault threshold $\alpha$ is then determined as the maximum reconstruction





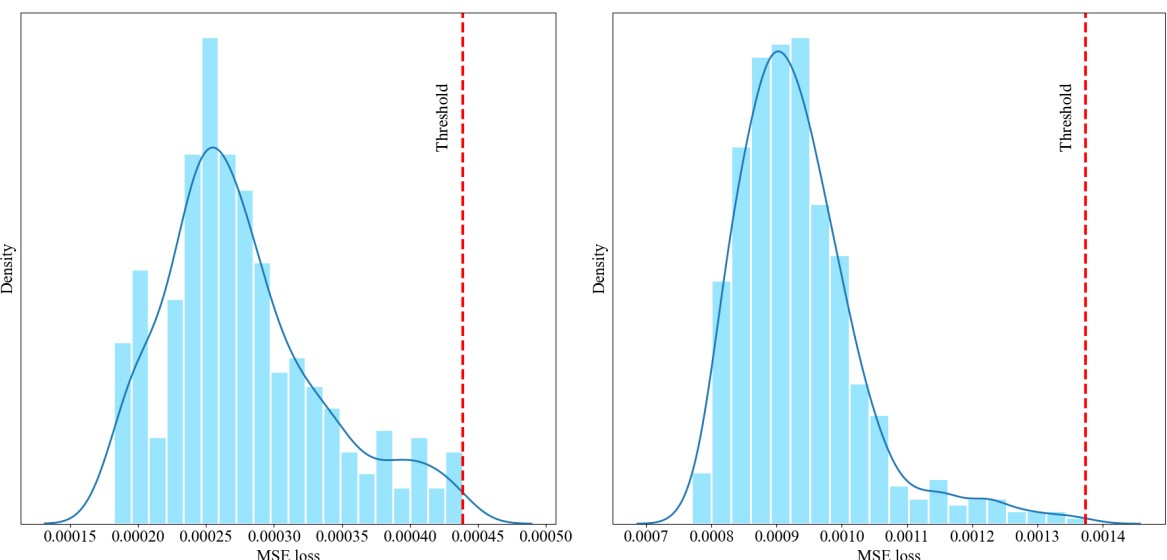

**Figure 8.** Histogram of anomaly scores for training data and threshold determination for two different signals: (a) SFt5, (b) AFtt

error observed in the training data as follows:

$$\alpha = \max_{i=1}^{N} e^{(i)} \tag{4}$$

where N is the size of the training dataset. As an example, Figures 8 (a) and (b) show the histogram of reconstruction errors or
anomaly scores for training (healthy) data and the fault threshold determination for two distinct measurements. These figures
clearly demonstrate that the thresholds are data-driven, varying in accordance with the input measurements of the CAE model.
Any test sample with a reconstruction error exceeding the threshold would be considered an anomaly.

### 3.3 Overall procedure of anomaly detection method

The steps taken in the anomaly detection of the PMSG model are briefly described in this section and outlined in Figure 9. The
methodology includes several stages as follows:

1. Data collection: The initial step involves the comprehensive gathering of target measurements, introduced in Table 2
   from the PMSG model.

2. Data preprocessing: Once collected, the measurements are preprocessed. As the first step, the signals are segmented into
   shorter-length signals to prepare training, validation, and test datasets for the CAE model. Segmented signals are then
   normalized to aid in efficient training, prevent numerical issues, and ultimately lead to better model performance and
   generalization. Both time-domain and frequency-domain measurements are fed into the model to compare the model
   performance in both cases for different measurements.





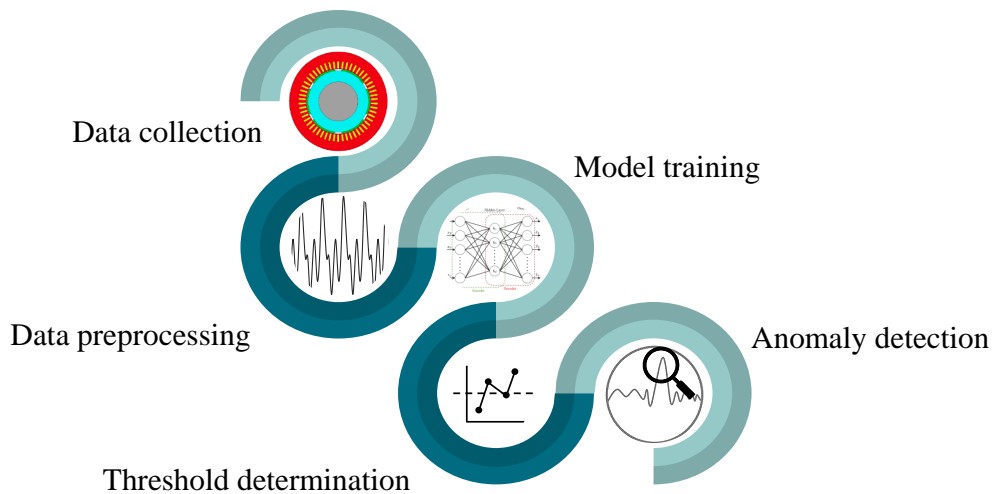

**Figure 9.** Overall procedure of anomaly detection method

3. Model training: With the data prepared, the next phase is the training of the CAE model. It should be noted that distinct CAE models are trained for different measurements, depending on the type of input data, either time-domain or
frequency-domain. The model learns to identify patterns and features of the input data representative of the normal operational state of the PMSG model.

4. Threshold determination: As discussed in Section 3.2, a key aspect of the methodology is establishing a reliable fault threshold. The optimal choice of this threshold is crucial for effectively differentiating between normal and faulty states and ensuring a precise alarm triggering. This study utilizes the maximum reconstruction error value from the training
dataset as the fault threshold.

5. Anomaly detection: The final step involves the actual detection of anomalies. The trained CAE model is employed to analyze new data, identifying deviations from the norm. Any reconstruction error exceeding the established fault threshold is considered indicative of an anomaly in the PMSG model.

### 3.4 Performance metrics

In this study, two performance metrics are used to evaluate the performance of the CAE model for anomaly detection:

– F1 score: This metric combines precision and recall to provide a single score for the model's overall accuracy in anomaly detection (Miele et al., 2022; Wang et al., 2019). Precision indicates what proportion of identified anomalies are true anomalies, and its equation is as follows:

$$\text{precision} = \frac{\text{TP}}{\text{TP} + \text{FP}} \tag{5}$$





where TP is the number of correctly identified anomalies (true positives), and FP is the number of misclassified normal samples as anomalies (false positives). Recall also specifies what proportion of true anomalies are identified and is determined as follows:

$$recall = \frac{TP}{TP + FN} \tag{6}$$

where FN is the number of misclassified anomaly samples as normal (false negative). F1 score ranges from 0 to 1, where 1 represents perfect precision and recall, and is defined as:

$$F1\ score = 2 \times \frac{precision \times recall}{precision + recall} \tag{7}$$

– Silhouette coefficient: This metric assesses the quality of clustering in unsupervised machine learning tasks, with scores ranging from -1 to 1 (Rousseeuw, 1987). In this study, the Silhouette score is used to calculate the average distance between the cluster of identified anomalies and the cluster of healthy data. A higher Silhouette coefficient indicates better separation between normal and anomaly clusters. The Silhouette coefficient for each sample $i$ is calculated as follows:

$$SC^{(i)} = \frac{b^{(i)} - a^{(i)}}{\max[a^{(i)}, b^{(i)}]} \tag{8}$$

where $a^{(i)}$ is the average distance between sample $i$ and all other samples in the same cluster. $b^{(i)}$ is also the minimum distance between sample $i$ and all samples in another cluster, not containing sample $i$.

Both metrics provide comprehensive insights into the performance and reliability of the CAE model in detecting anomalies using different measurement variables.

## 4 Results and discussion

### 4.1 Data preparation

The procedure for anomaly detection begins with collecting simulated measurements from the PMSG model under both healthy and various faulty conditions. These measurements are then divided into shorter-length signals to ensure an adequate amount of training and test data for the CAE model. To compare the performance of different measurements, both time-domain data (raw segmented signals) and frequency-domain data (spectrum of segmented signals) are inputted into the CAE model. Following the standard approach for anomaly detection, the training dataset consists only of measurements from the healthy state, while the test dataset encompasses all states, including healthy and different faulty conditions. The original signals are sampled at a rate of $F_s = 21.132$ kHz. The fundamental frequency of the generator model under study is calculated as $f = (N_s \times P)/120 = 58.69$ Hz, where $N_s$ represents the synchronous speed of the generator at its rated speed of 1173.7 RPM, and $P$ denotes the number of poles (six in this case).


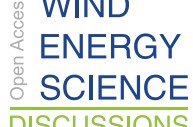

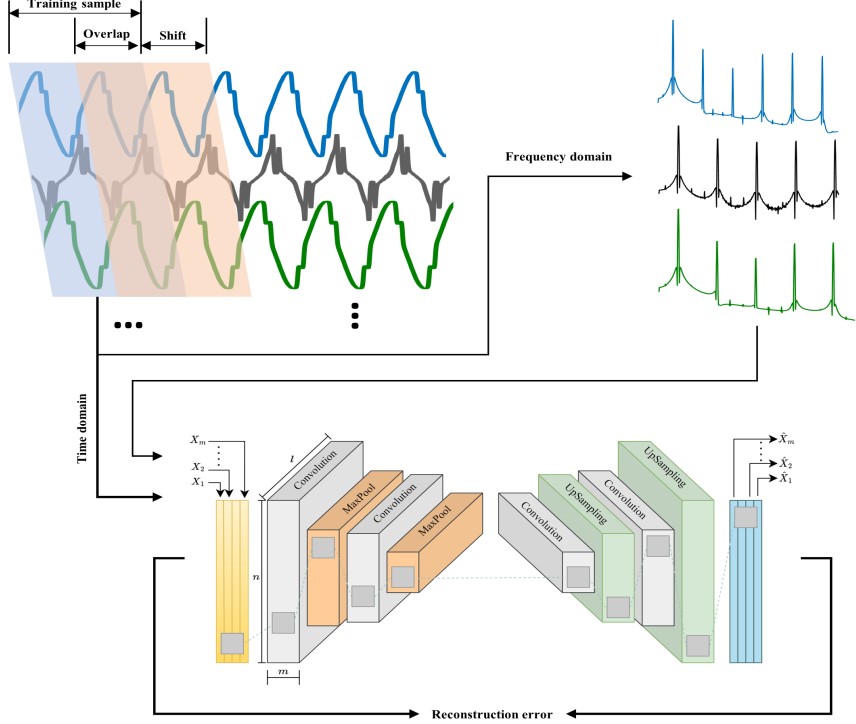

**Figure 10.** Signal segmentation process

For time-domain data inputted into the CAE model, the length of segmented signals is set to twice the period of the fundamental frequency, resulting in 720 sample points per segment. Additionally, the segmentation process includes overlap between consecutive segments equivalent to one period of the fundamental frequency, which is 360 sample points. It's important to highlight that for frequency-domain data, where a higher resolution in obtaining the frequency spectrum is required, the segmented signals have a longer duration of 30 times the period of the fundamental frequency or 10800 sample points. This longer duration leads to smaller training and test datasets for the frequency-domain cases compared to the time-domain cases. It should be noted that only the first 720 sample points of the frequency spectrum are fed into the model in order to have the same CAE model architecture for time-domain and frequency-domain cases. The segmentation and feeding procedure into the CAE model for both time-domain and frequency-domain cases is illustrated in Figure 10.

## 4.2 Anomaly detection results: time domain inputs

The CAE model's reconstruction errors or anomaly scores trained with time-domain data for demagnetization and eccentricity faults are illustrated in Figures 11 and 12, respectively. These figures highlight the reconstruction error values within a yellow region for the set of healthy training samples, where the highest error value defines the fault threshold, marked by a horizontal dotted red line. Similarly, the green region represents the reconstruction errors for healthy test samples. The outcomes of





performance metrics are detailed in Table 5. According to the F1 score results, nearly all signals can train a CAE model that effectively detects demagnetization fault cases. In contrast, for eccentricity fault cases, only specific signals—induced shaft voltage ($Vsh$), electromagnetic torque ($Te$), stator phase currents ($Is$), and the tangential component of airgap flux at tooth positions ($AFtt$)—achieve full accuracy in fault detection. The Silhouette coefficient, which measures the degree of separation between anomaly clusters and the healthy cluster, shows that $Vsh$, $Te$, $Is$, and $AFtt$ signals provide superior separation for both demagnetization and eccentricity faults.

**Table 5.** Performance metric results of anomaly detection models trained with time-domain data

| Input variable | Demagnetization | | Eccentricity | |
| --- | --- | --- | --- | --- |
| | F1 | Silhouette | F1 | Silhouette |
| $Vsh$ | 1.0000 | 0.9863 | 1.0000 | 0.9890 |
| $Te$ | 1.0000 | 0.9996 | 1.0000 | 0.9998 |
| $Is$ (single phase) | 0.9993 | 0.9999 | 0.9993 | 0.9998 |
| $Is$ (three phases) | 1.0000 | 0.9999 | 1.0000 | 0.9999 |
| $SFr5$ | 0.9980 | 0.8592 | 0.0650 | 0.1690 |
| $SFr10$ | 1.0000 | 0.8600 | 0.0690 | 0.0840 |
| $SFt5$ | 1.0000 | 0.8990 | 0.3000 | 0.2400 |
| $SFt10$ | 1.0000 | 0.9100 | 0.3400 | 0.3500 |
| $AFrt$ | 1.0000 | 0.9500 | 0.8700 | 0.7800 |
| $AFrs$ | 1.0000 | 0.9400 | 0.5100 | 0.4900 |
| $AFtt$ | 1.0000 | 0.9721 | 1.0000 | 0.9835 |
| $AFts$ | 0.9300 | 0.8600 | 0.8000 | 0.6600 |

The sensitivity of the CAE model to fault severity is also evaluated using the trend of anomaly score values for fault scenarios. It is expected that cases with higher fault severity should correspond to higher anomaly scores. Among the models trained with the signals featured, the model trained with three-phase $Is$ signals demonstrates sensitivity to the severity across all types of demagnetization faults, as illustrated in Figure 11 (d). However, the $Vsh$-based trained model does not adequately reflect the severity for FC3 and FC4 cases, which are localized demagnetization in one pole, with the anomaly scores for FC4 unexpectedly lower. Other models, including those trained with $Te$), stray, and airgap flux signals, fail to differentiate between the severities of FC5 and FC6 cases, both involving localized demagnetization across all poles.

For eccentricity faults, both dynamic (FC1-FC3) and static (FC4-FC6), as presented in Figure 12, the $Is$ and $Vsh$ signals are effective in distinguishing between different degrees of eccentricity for both cases, with anomaly scores increasing as the degree of eccentricity raises. However, outputs of the CAE model trained with $Te$ signal do not display a consistent trend with respect to the eccentricity degree. The results for FC2 test samples, which exhibit a 15% dynamic eccentricity, are also poor when the model is trained with $AFtt$ signals.



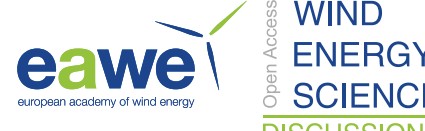

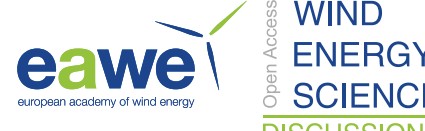

**Figure 11.** Anomaly detection results of time-domain measurements for demagnetization fault cases: (a) $Vsh$, (b) $Te$, (c) $Is$ (single phase), (d) $Is$ (three phase), (e) $SFr5$, (f) $SFr10$, (g) $SFt5$, (h) $SFt10$, (i) $AFrt$, (j) $AFrs$, (k) $AFtt$, and (l) $AFts$

**Figure 12.** Anomaly detection results of time-domain measurements for static and dynamic eccentricity fault cases: (a) $Vsh$, (b) $Te$, (c) $Is$ (single phase), (d) $Is$ (three phase), (e) $SFr5$, (f) $SFr10$, (g) $SFt5$, (h) $SFt10$, (i) $AFrt$, (j) $AFrs$, (k) $AFtt$, and (l) $AFts$

In summary, the CAE model, trained with time-domain data from three-phase stator currents, reliably provides anomaly detection for both demagnetization and eccentricity fault cases. The efficacy of this model with frequency domain data will be further assessed in subsequent analyses.





### 4.3 Anomaly detection results: frequency domain inputs

In this section, segmented signals are transformed into the frequency domain. This transformation allows for the frequency
information of simulated data to be utilized by the CAE model, thereby enhancing its predictive accuracy. The use of frequency-
domain information has been found to improve prediction outcomes for certain signals. To illustrate this, the power spectral
density (PSD) spectra for a variety of signals are shown in Figure 13. These include healthy data, demagnetization in FC5
and FC6 (two segments of all poles), and static eccentricity (FC4-FC6). Upon examination of these figures, it is observed that
fluctuations in frequency content associated with faults, as well as the appearance of characteristic fault frequencies, are more
prominently visible in specific signals, notably those from flux sensors and electromagnetic torque.

Similar to the anomaly detection results in the time domain, the anomaly scores for frequency-domain training and test
samples, utilizing various models, are depicted in Figures 14 and 15 for demagnetization and eccentricity faults, respectively.
Additionally, the performance of these models is evaluated and presented in terms of F1 and Silhouette scores, as shown in
Table 6.

**Table 6.** Performance metric results of anomaly detection models trained with frequency-domain data

| Input variable | Demagnetization | | Eccentricity | |
|---|---|---|---|---|
| | F1 | Silhouette | F1 | Silhouette |
| $Vsh$ | 1.00 | 0.9912 | 0.9733 | 0.9321 |
| $Te$ | 1.00 | 0.9999 | 1.0000 | 0.9922 |
| $Is$ (single phase) | - | - | 0.8623 | 0.7760 |
| $Is$ (three phases) | - | - | 0.8230 | 0.6561 |
| $SFr5$ | 1.00 | 0.9220 | 0.9692 | 0.7914 |
| $SFr10$ | 1.00 | 0.9190 | 0.7300 | 0.8580 |
| $SFt5$ | 1.00 | 0.9320 | 1.0000 | 0.6943 |
| $SFt10$ | 1.00 | 0.9500 | 1.0000 | 0.700 |
| $AFrt$ | 0.91 | 0.9620 | 1.0000 | 0.9400 |
| $AFrs$ | 0.91 | 0.9670 | 0.6900 | 0.9400 |
| $AFtt$ | 0.91 | 0.9845 | 1.0000 | 0.9899 |
| $AFts$ | 0.91 | 0.9950 | 0.7000 | 0.9400 |

The CAE model, when trained with the $Vsh$ signal, demonstrates reliable anomaly detection results in demagnetization
cases, as depicted in Figure 14 (a), highlighting both detection accuracy and sensitivity to fault severity. However, in the
context of eccentricity faults, this model exhibits somewhat poorer performance in detecting FC3 (25% dynamic eccentricity)
and in being sensitive to different levels of fault severity, as illustrated in Figure 15) (a).



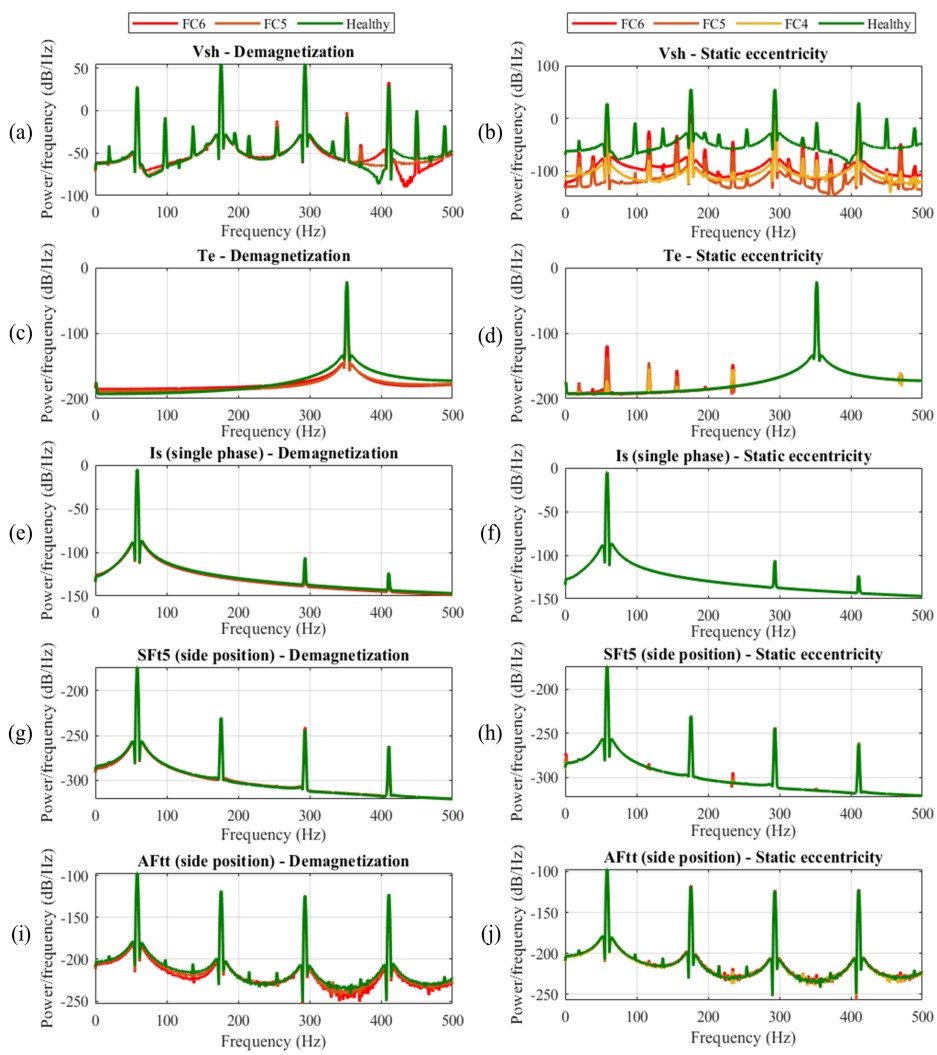

**Figure 13.** Power spectral density plots of simulated signals under healthy and faulty states

While the $Te$ signal achieves full detection accuracy for both types of faults according to Table 6, similar to the results in
the time domain, the anomaly scores do not align with the actual severity levels of the fault cases. This discrepancy is evident
in the results depicted in Figures 14 (b) and 15 (b).

Stator current signals ($Is$) yield anomaly scores that are entirely below the fault threshold line for demagnetization cases, and
partially below for eccentricity cases, as illustrated in Figures 14 and 15 (c) and (d). This outcome aligns with expectations,
as the comparison of the frequency spectra of the $Is$ signal between healthy and faulty states, shown in Figures 13 (e) and
(f), reveals only minor shifts attributable to the faults. Furthermore, it is noteworthy that, according to Figures 14 and 15 (c)
and (d), the CAE model tends to assign lower anomaly scores to faulty samples than to healthy ones. This issue is attributed



**Figure 14.** Anomaly detection results of frequency-domain measurements for demagnetization fault cases: (a) $Vsh$, (b) $Te$, (c) $Is$ (single phase), (d) $Is$ (three phase), (e) $SFr5$, (f) $SFr10$, (g) $SFt5$, (h) $SFt10$, (i) $AFrt$, (j) $AFrs$, (k) $AFtt$, and (l) $AFts$



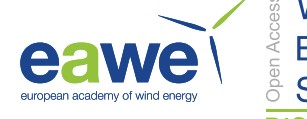

**Figure 15.** Anomaly detection results of frequency-domain measurements for static and dynamic eccentricity fault cases: (a) $Vsh$, (b) $Te$, (c) $Is$ (single phase), (d) $Is$ (three phase), (e) $SFr5$, (f) $SFr10$, (g) $SFt5$, (h) $SFt10$, (i) $AFrt$, (j) $AFrs$, (k) $AFtt$, and (l) $AFts$





to the subtle feature shifts caused by faults and the CAE model's complexity, which enables it to capture a broad spectrum of features, including those not directly indicative of anomalies. The model's high complexity, advantageous for identifying complex patterns in normal data, might also unexpectedly improve its ability to reconstruct faulty samples. This is because the model, with its extensive layers and multitude of parameters, can generalize well to data variations that resemble the healthy samples it was trained on, even if they are not identical. Consequently, subtle feature shifts due to faults are not adequately penalized, resulting in lower anomaly scores for faulty samples. This phenomenon is not unique to stator current signals; it has also been observed in other scenarios, such as the FC5 demagnetization case with airgap flux sensors, as depicted in Figures 14 (i)-(l). Addressing this issue requires a comprehensive understanding of model complexity and its effects on anomaly detection, which is beyond the scope of this study.

Stray flux sensors, both radial and tangential components, demonstrate strong performance by achieving full accuracy and sensitivity in demagnetization cases, as evidenced in Figures 14 (e)-(k). In the context of eccentricity faults, the radial components ($SFr5$ and $SFr10$) exhibit effective detection of dynamic eccentricity (FC1-FC3) test samples, with the severity levels being distinctly identifiable. However, these signals, particularly $SFr10$, fail to detect some test samples associated with static eccentricity (FC4-FC6), as reflected by the F1 scores (96.92% for $SFr5$ and 73% for $SFr10$) in Table 6 and Figures 15 (e) and (f). On the other hand, tangential component signals ($SFt5$ and $SFt10$) achieve full F1 scores, and the models trained on these signals are sensitive to varying severity levels of both dynamic and static eccentricities, as shown in Figures 15) (g) and (h). Nevertheless, based on Silhouette scores, it is observed that radial components offer a superior separation between healthy and faulty clusters in cases of eccentricity faults.

Regarding airgap flux sensors, the anomaly scores related to demagnetization, as presented in Figures 14 (i)-(l), reveal an inability to detect test samples for FC5, which represents a low-severity, localized demagnetization affecting all poles. Furthermore, both the radial and tangential components of the airgap flux sensors positioned at the slot ($AFrs$ and $AFts$) fail to detect static eccentricity cases, as indicated in Figures 15 (j) and (l). However, the tooth sensor, especially its tangential component ($AFtt$), gives precise results for eccentricity cases that accurately reflect the severity levels of the faults, as demonstrated in Figure 15 (k).

In summary, CAE models trained on the frequency content from stray flux sensors at all four top, bottom, and side positions - tangential components in both 5mm and 10mm distance, and radial components in 5mm distance outside the stator housing - demonstrate full accuracy and sensitivity in detecting anomalies associated with both demagnetization and eccentricity faults. Moreover, the radial and tangential components of airgap flux sensors located at the tooth are fully capable of identifying all types of eccentricity anomalies, accurately reflecting their severity levels. These findings highlight the potential of adopting flux monitoring techniques—well-established in other industries for their cost-efficient and easy-to-install sensors—in the fault detection of large MW offshore wind generators. The utilization of such sensors, especially stray flux sensors enhanced with frequency information, presents a promising strategy for the condition monitoring of these systems.

To conclude the discussion section, Table 7 provides a comprehensive and detailed summary of all anomaly detection outcomes derived from the previously analyzed signals. This table outlines the detection accuracy for each fault case, alongside the sensitivity to fault severity, across both time and frequency domains. The table is organized such that the first row associated





with each domain presents the detection accuracy results, while the second row specifies whether the signal in question is capable of monitoring changes in fault severity levels. It should be noted that for the purposes of this table, detection accuracies exceeding 95% are considered to constitute acceptable accuracy. This structured presentation ensures a clear and concise overview of the study's findings.

Table 7: Summary of anomaly detection results

| Signal | Domain | Demagnetization |||||| Eccentricity ||||||
|---|---|---|---|---|---|---|---|---|---|---|---|---|---|
| | | FC1 | FC2 | FC3 | FC4 | FC5 | FC6 | FC1 | FC2 | FC3 | FC4 | FC5 | FC6 |
| $Vsh$ | Time | ✓ | ✓ | ✓ | ✓ | ✓ | ✓ | ✓ | ✓ | ✓ | ✓ | ✓ | ✓ |
| | | ✓ | | ✗ | | ✓ | | ✓ | | | ✓ | | |
| | Freq. | ✓ | ✓ | ✓ | ✓ | ✓ | ✓ | ✓ | ✓ | ✓ | ✓ | ✓ | ✓ |
| | | ✓ | | ✓ | | ✓ | | ✗ | | | ✗ | | |
| $Te$ | Time | ✓ | ✓ | ✓ | ✓ | ✓ | ✓ | ✓ | ✓ | ✓ | ✓ | ✓ | ✓ |
| | | ✓ | | ✓ | | ✗ | | ✗ | | | ✗ | | |
| | Freq. | ✓ | ✓ | ✓ | ✓ | ✓ | ✓ | ✓ | ✓ | ✓ | ✓ | ✓ | ✓ |
| | | ✗ | | ✗ | | ✓ | | ✗ | | | ✗ | | |
| $Is$ - single phase | Time | ✓ | ✓ | ✓ | ✓ | ✓ | ✓ | ✓ | ✓ | ✓ | ✓ | ✓ | ✓ |
| | | ✗ | | ✗ | | ✓ | | ✓ | | | ✓ | | |
| | Freq. | ✗ | ✗ | ✗ | ✗ | ✗ | ✗ | ✓ | ✓ | ✗ | ✓ | ✓ | ✗ |
| | | ✗ | | ✗ | | ✗ | | ✗ | | | ✗ | | |
| $Is$ - three phases | Time | ✓ | ✓ | ✓ | ✓ | ✓ | ✓ | ✓ | ✓ | ✓ | ✓ | ✓ | ✓ |
| | | ✓ | | ✓ | | ✓ | | ✓ | | | ✓ | | |
| | Freq. | ✗ | ✗ | ✗ | ✗ | ✗ | ✗ | ✓ | ✗ | ✗ | ✓ | ✓ | ✗ |
| | | ✗ | | ✗ | | ✗ | | ✗ | | | ✗ | | |
| $SFr5$ | Time | ✓ | ✓ | ✓ | ✓ | ✓ | ✓ | ✗ | ✗ | ✗ | ✗ | ✗ | ✗ |
| | | ✓ | | ✓ | | ✗ | | ✗ | | | ✗ | | |
| | Freq. | ✓ | ✓ | ✓ | ✓ | ✓ | ✓ | ✓ | ✓ | ✓ | ✓ | ✓ | ✓ |
| | | ✓ | | ✓ | | ✓ | | ✓ | | | ✓ | | |
| $SFt5$ | Time | ✓ | ✓ | ✓ | ✓ | ✓ | ✓ | ✗ | ✗ | ✗ | ✗ | ✗ | ✗ |
| | | ✓ | | ✓ | | ✗ | | ✗ | | | ✗ | | |

 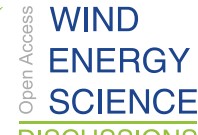

**Table 7 continued from previous page**

| Signal | Domain | Demagnetization | | | | | | Eccentricity | | | | | |
|---|---|---|---|---|---|---|---|---|---|---|---|---|---|
| | | FC1 | FC2 | FC3 | FC4 | FC5 | FC6 | FC1 | FC2 | FC3 | FC4 | FC5 | FC6 |
| | Freq. | ✓ | ✓ | ✓ | ✓ | ✓ | ✓ | ✓ | ✓ | ✓ | ✓ | ✓ | ✓ |
| | | ✓ | | ✓ | | ✓ | | ✓ | | | ✓ | | |
| $SFr10$ | Time | ✓ | ✓ | ✓ | ✓ | ✓ | ✓ | ✗ | ✗ | ✗ | ✗ | ✗ | ✗ |
| | | ✓ | | ✓ | | ✗ | | ✗ | | | ✗ | | |
| | Freq. | ✓ | ✓ | ✓ | ✓ | ✓ | ✓ | ✓ | ✓ | ✓ | ✗ | ✗ | ✗ |
| | | ✓ | | ✓ | | ✓ | | ✓ | | | ✗ | | |
| $SFt10$ | Time | ✓ | ✓ | ✓ | ✓ | ✓ | ✓ | ✗ | ✗ | ✗ | ✗ | ✗ | ✗ |
| | | ✓ | | ✓ | | ✗ | | ✗ | | | ✗ | | |
| | Freq. | ✓ | ✓ | ✓ | ✓ | ✓ | ✓ | ✓ | ✓ | ✓ | ✓ | ✓ | ✓ |
| | | ✓ | | ✓ | | ✓ | | ✓ | | | ✓ | | |
| $AFrt$ | Time | ✓ | ✓ | ✓ | ✓ | ✓ | ✓ | ✗ | ✓ | ✓ | ✗ | ✓ | ✓ |
| | | ✓ | | ✓ | | ✗ | | ✓ | | | ✓ | | |
| | Freq. | ✓ | ✓ | ✓ | ✓ | ✗ | ✓ | ✓ | ✓ | ✓ | ✓ | ✓ | ✓ |
| | | ✓ | | ✓ | | ✓ | | ✓ | | | ✓ | | |
| $AFtt$ | Time | ✓ | ✓ | ✓ | ✓ | ✓ | ✓ | ✓ | ✓ | ✓ | ✓ | ✓ | ✓ |
| | | ✓ | | ✓ | | ✗ | | ✗ | | | ✓ | | |
| | Freq. | ✓ | ✓ | ✓ | ✓ | ✗ | ✓ | ✓ | ✓ | ✓ | ✓ | ✓ | ✓ |
| | | ✓ | | ✓ | | ✓ | | ✓ | | | ✓ | | |
| $AFrs$ | Time | ✓ | ✓ | ✓ | ✓ | ✓ | ✓ | ✗ | ✗ | ✓ | ✗ | ✗ | ✓ |
| | | ✓ | | ✓ | | ✗ | | ✓ | | | ✓ | | |
| | Freq. | ✓ | ✓ | ✓ | ✓ | ✗ | ✓ | ✓ | ✓ | ✓ | ✗ | ✗ | ✗ |
| | | ✓ | | ✓ | | ✓ | | ✓ | | | ✗ | | |
| $AFts$ | Time | ✓ | ✓ | ✓ | ✓ | ✗ | ✓ | ✗ | ✓ | ✓ | ✗ | ✓ | ✓ |
| | | ✓ | | ✓ | | ✓ | | ✓ | | | ✓ | | |
| | Freq. | ✓ | ✓ | ✓ | ✓ | ✗ | ✓ | ✓ | ✓ | ✓ | ✗ | ✗ | ✗ |
| | | ✓ | | ✓ | | ✓ | | ✓ | | | ✗ | | |

Abbreviations: $Vsh$, Induced shaft voltage; $Te$, Electromagnetic torque; $Is$, Stator phase current; $SFr5$, Stray flux sensor - radial component, outside stator housing with a distance of 5mm (4 signals); $SFt5$, Stray flux sensor - tangential component, outside stator housing with a distance





of 5mm (4 signals); $SFr10$, Stray flux sensor - radial component, outside stator housing with a distance of 10mm (4 signals); $SFt10$, Stray flux sensor - tangential component, outside stator housing with a distance of 10mm (4 signals); $AFrt$, Airgap flux sensor - radial component at tooth

position (4 signals); $AFtt$, Airgap flux sensor - tangential component at tooth position (4 signals); $AFrs$, Airgap flux sensor - radial component at slot position (4 signals); $AFts$, Airgap flux sensor - tangential component at slot position (4 signals); Demagnetization: FC1, One pole all segments 10%; FC2, One pole all segments 20%; FC3, 2 segments of one pole 40% & 20%; FC4, 2 segments of one pole 80% & 40%; FC5, 2 segments of all poles 20% & 10%; FC6, 2 segments of all poles 40% & 20%; Eccentricity: FC1-FC3, Dynamic 5-25%; FC4-FC6, Static 5-25%.

### 4.4    Fault discrimination capability of selected measurements

This section evaluates the performance of the CAE anomaly detection model, focusing on its fault discrimination capability. The analysis covers models trained on selected signals from both the time and frequency domains, as discussed in earlier sections and presented in Table 7. The CAE model, trained with time-domain signals of three-phase current $Is$, achieved perfect accuracy in detecting anomalies and showed sensitivity to variations in fault severities, as depicted in Figures 11 (d) and 12 (d). In the frequency domain, models trained with stray flux signals - tangential components in both 5 mm ($SFt5$)

and 10 mm ($SFt10$) distances and radial components in 5 mm distance ($SFr5$) - proved to be more effective in detecting anomalies than others.

    The ability of these models to distinguish between two specific types of faults is assessed by analyzing the reconstruction errors. This analysis is presented in Figures 16 (a)-(d) for $Is$, $SFt5$, $SFt10$, and $SFr5$ signals, respectively. While $Is$ signals allow for perfect differentiation between different fault cases, there is a noticeable overlap in anomaly score ranges between

eccentricity and demagnetization faults, as shown in Figure 16 (a). This overlap makes establishing a clear separation boundary challenging between eccentricity and demagnetization, as anomaly scores for FC1 and FC3 demagnetization faults are lower than those for eccentricity faults. A similar issue is observed with the tangential stray flux ($SFt5$ and $SFt10$) measurements, as indicated in Figures 16 (b) and (c). However, the CAE model trained with radial stray flux ($SFr5$) measurements demonstrates a clear ability to differentiate between the two fault types based on their anomaly scores, as shown in Figure 16 (c).

These findings indicate that radial stray flux measurements, taken from 5 mm outside the stator housing, not only accurately detect anomalies and assess fault severity but also effectively distinguish between eccentricity and demagnetization faults in PMSG. Further exploration of fault classification and diagnosis will be the focus of future work.

### 5    Conclusions

This study has successfully demonstrated the application of convolutional autoencoder (CAE) models for anomaly detection

in offshore wind permanent magnet synchronous generators (PMSGs), addressing demagnetization and eccentricity faults of varying severity. Utilizing a simulation high-speed PMSG model designed based on the specifications of the NREL 5-MW reference offshore wind turbine, this research employed unsupervised CAE models trained on healthy state simulation data to analyze a range of signals, including three-phase currents, induced shaft voltage, electromagnetic torque, and airgap and stray magnetic flux. While some of these measurements, such as phase currents, are typically included in the Supervisory

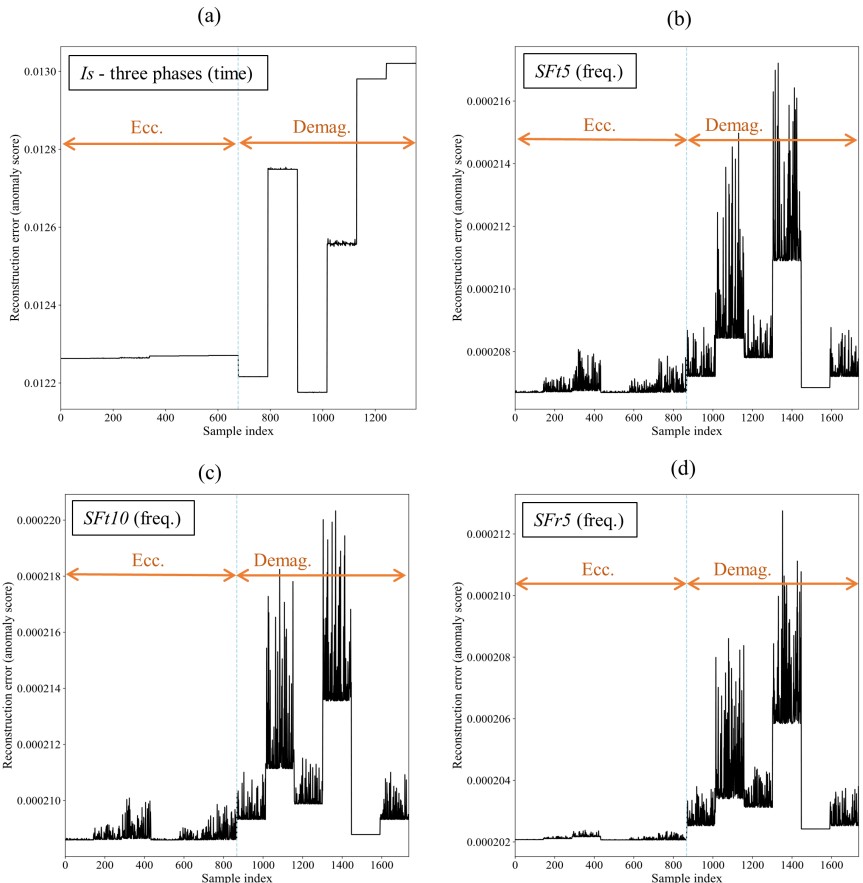

**Figure 16.** Anomaly detection results for both eccentricity and demagnetization: (a) time domain $Is$ (three phases), (b) frequency domain $SFt5$, (c) frequency domain $SFt10$, and (d) frequency domain $SFs5$.

385 Control and Data Acquisition (SCADA) systems of wind turbines, they are often at a low resolution. A key limitation of this downsampled data is its inability to detect emerging failures in PMSGs as promptly as necessary. Hence, this study aimed to evaluate the potential of high-resolution measurements for the early detection of possible failures in PMSGs. The findings indicate that three-phase currents in the time domain, along with a combination of top, bottom, and side positions of stray flux sensors—both tangential and radial components in the frequency domain—significantly enhance anomaly detection accuracy

390 and fault severity sensitivity. Notably, the radial components of stray flux sensors proved capable of differentiating between types of eccentricity and demagnetization faults. The findings suggest that using flux monitoring techniques, with cost-efficient and easily installed stray flux sensors with frequency information, could be an effective strategy for early fault detection in large MW offshore wind generators. Future work will focus on further validating these results with experimental data and exploring



the impact of varying measurement resolutions to determine the minimum resolution necessary for early fault detection, thereby

confirming the models' effectiveness in practical scenarios.

*Code availability.* The code used in this study is available upon request.

*Author contributions.* **Ali Dibaj**: Writing – original draft, Writing – review & editing, Validation, Software, Methodology, Investigation, Formal analysis, Conceptualization. **Mostafa Valavi**: Final review & editing, Data curation, Conceptualization, Formal analysis, Funding acquisition. **Amir R. Nejad**: Final review & editing, Supervision, Formal analysis, Project administration, Funding acquisition.

*Competing interests.* Amir R. Nejad is a member of the editorial board of Wind Energy Science. The authors declare that they have no other known competing financial interests or personal relationships that could have appeared to influence the work reported in this paper.

*Acknowledgements.* The authors gratefully acknowledge the financial support of the Research Council of Norway through the InteDiag-WTCP project (Project Number 309205).





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
