# Peer review of "Unsupervised anomaly detection of permanent magnet offshore wind generators through electrical and electromagnetic measurements"

_Wind Energy Science, 2024_

## Author Comment (AC1)

Dear Handling Associate Editor,

Thank you for your message regarding our submitted paper entitled:

**Unsupervised Anomaly Detection of Permanent Magnet Offshore Wind Generators Through Electrical and Electromagnetic Measurements**

and for the opportunity to revise it. We also appreciate the reviewers' comments, which were very helpful in revising our manuscript. We have carefully addressed each point raised by the reviewers, and corresponding revisions have been made to our paper. Below are our responses to the comments provided:

**Responses to Reviewer #1 comments (RC1):**

The manuscript provides a method to detect demagnetization and eccentricity in a permanent magnet synchronous generator using an unsupervised convolution autoencoder model. The method uses measured signals to detect the fault. The manuscript effectively outlines its objective and hypotheses. However, it is recommended that the authors address the following comments to enhance the clarity of the manuscript:

**Comment#1:**

Provide a comprehensive literature survey on the fault detection method using machine learning and compare it with the proposed method.

**Answer#1:**

We appreciate the reviewer's insightful comment. The revised manuscript now includes an expanded literature review on recent machine learning-based fault detection methods for electrical machines, specifically focusing on permanent magnet synchronous machines. For instance, Nguyen et al. [1] implemented a gradient boosting machine for detecting inter-turn short-circuit faults and local demagnetization using current and stray flux measurements. Cai et al. [2] utilized vibration and acoustic emission data with a combined complementary ensemble empirical model decomposition and Bayesian network model for fault detection in rolling element bearings. Huang et al. [3] developed a semi-supervised rule-based classifier for demagnetization fault diagnosis, while Tan et al. [4] explored the use of current measurements combined with an artificial neural network (ANN) for detecting faults in converter systems of PMSGs. Penrose [5] investigated the application of k-Nearest Neighbors (KNN) for fault classification and linear regression models for estimating RUL, providing a 30-day advance notification of failures in small electric machines using basic data inputs. Ehya et al. [6] examined different machine learning and signal processing techniques for diagnosing inter-turn short-circuit faults in salient pole synchronous generators.

The literature reveals extensive application of machine learning in fault detection across general electrical machines. However, most recent studies focus on developing new machine learning models rather than optimizing them for specific applications, such as exploring the effectiveness of various measurements. In contrast, our study does not aim to develop a new machine learning method but adapts the presented convolutional autoencoder (CAE) anomaly detection model with optimized measurements for large MW wind generators, a field still ripe for research, particularly regarding early fault detection using machine learning.

In response to your comment about comparing the CAE model used in this study with other techniques, the revised manuscript now includes a comparison of three distinct models for unsupervised anomaly detection: One-Class Support Vector Machine (SVM), K-Nearest Neighbors (KNN), and K-Means. All models were trained on a dataset

of 400 normal samples using time-domain data of three-phase current measurements and tested using 100 untrained normal samples and 100 samples under demagnetization and eccentricity fault conditions. The One-Class SVM, employing an RBF kernel, focuses on defining a decision function that envelops the region of normal data, treating all other areas as anomalies. The K-Means model, configured with a single cluster, detects anomalies based on the distance from data points to the centroid of normal data, with a threshold set at the 99th percentile of these distances. The KNN model, using 10 neighbors, identifies anomalies based on the average distance to the nearest neighbors, applying a threshold at the 99th percentile to define outliers. Each model's performance was evaluated based on their F1 score and accuracy, with results summarized in Table 1 and illustrated through confusion matrices in Figure 1. The CAE model achieved the highest accuracy at 100%, while the compared traditional models recorded accuracies between 85-90%. It should be noted that the performance of these traditional models may diminish as data complexity increases, highlighting challenges in scaling these models to more complex or larger datasets. On the other side, it should also be noted that, the CAE deep learning model, requires extensive fine-tuning of numerous hyperparameters and significantly longer training times.

These comparison results have been added as a separate subsection in Section 4 of the revised manuscript.

Table 1: Comparison of performance metrics for anomaly detection models

| Metric | K-Means | KNN | One-Class SVM | CAE (proposed) |
|---|---|---|---|---|
| F1 Score | 0.8361 | 0.8672 | 0.8878 | 1.0000 |
| Accuracy (%) | 85.5 | 85.0 | 88.0 | 100.0 |

[Figure]

Figure 1: Confusion matrices for performance evaluation across different anomaly detection models trained with time-domain data of three-phase stator current measurements

[1]     N. Du Hoang Nguyen, V. K. Huynh, and K. G. Robbersmyr, "Current and Stray Flux Sensors for Anomaly Detection in PMSM Drive Based on Gradient Boosting Machine," *Proc. IEEE Sensors*, pp. 1–4, 2023, doi: 10.1109/SENSORS56945.2023.10325076.

[2]     B. Cai *et al.*, "Data-driven early fault diagnostic methodology of permanent magnet synchronous motor," *Expert Syst. Appl.*, vol. 177, no. February, p. 115000, 2021, doi: 10.1016/j.eswa.2021.115000.

[3]     F. Huang *et al.*, "Demagnetization Fault Diagnosis of Permanent Magnet Synchronous Motors Using Magnetic Leakage Signals," *IEEE Trans. Ind. Informatics*, vol. 19, no. 4, pp. 6105–6116, 2023, doi: 10.1109/TII.2022.3165283.

[4]     Y. Tan, H. Zhang, and Y. Zhou, "Fault Detection Method for Permanent Magnet Synchronous Generator Wind Energy Converters Using Correlation Features among Three-phase Currents," *J. Mod. Power Syst. Clean Energy*, vol. 8, no. 1, pp. 168–178, 2020, doi: 10.35833/MPCE.2018.000347.

[5]     H. W. Penrose, "Machine Learning for Electric Machine Prognostics and Remaining Useful Life with Basic Motor Data," *2022 IEEE Electr. Insul. Conf. EIC 2022*, vol. 2022-Janua, no. June, pp. 245–248, 2022, doi: 10.1109/EIC51169.2022.10122613.

[6]     H. Ehya, T. N. Skreien, and A. Nysveen, "Intelligent Data-Driven Diagnosis of Incipient Interturn Short Circuit Fault in Field Winding of Salient Pole Synchronous Generators," *IEEE Trans. Ind. Informatics*, vol. 18, no. 5, pp. 3286–3294, 2022, doi: 10.1109/TII.2021.3054674.

**Comment#2:**

How the shaft voltage in Fig. 4 (d) is simulated? Is the impact of the converter excitation and the bearing current considered during simulation?

**Answer#2:**

The shaft voltage illustrated in Figure 4(d) is simulated to result solely from an unbalanced magnetic field due to faults. We have not modeled the effects of converter-fed operation and bearing currents in our simulations, as these factors are outside the scope of this study. Future research may consider incorporating these elements to broaden the model's relevance to practical applications.

**Comment#3:**

How the eccentricity is simulated. Quantify the misalignment of the rotor relative to the stator?

**Answer#3:**

Eccentricity cases are modeled in the 2D (xy-plane) to represent the non-uniform airgap effectively. In static eccentricity, the stator axis is shifted while the rotor remains fixed at (0,0), creating a constant non-uniform airgap whose length distribution does not vary with rotation. In dynamic eccentricity, the rotor axis moves during rotation, maintaining its fixed position at (0,0) for the stator, which results in a varying airgap length distribution through the rotation cycle. The degree of eccentricity is quantified by the offset value of the shifted axis relative to the airgap length in the healthy condition.

**Comment#4:**

Fig. 13 (e-f) shows no change in the power spectral density of the phase current between the healthy condition and fault condition. Are the time-domain signals also same? If yes, then how it is used in section 4.2 to detect the fault condition?

**Answer#4:**

Thank you for your observation regarding the power spectral density (PSD) plots in Figures 13 (e) and (f). It is indeed correct that no significant changes are observed in these PSD spectrums of phase current between healthy and fault conditions. To clarify, the PSD plots from original phase current measurements for different conditions are illustrated in Figure 2. These plots show only subtle variations in amplitude due to faults, with minor deviations noted. This is partly because the PSD spectrums are based on raw signals of 0.5 seconds duration (~10,800 points), captured at a sample rate of 21.132 kHz, which might not offer sufficient frequency resolution to highlight fault-related deviations effectively.

Further analysis indicates that if the frequency resolution were improved, these deviations could be more distinctly observed. As outlined in the Conclusion Section, future research will focus on studying signal resolution and its impact on fault detection effectiveness.

[Figure]

Figure 2: Power spectral density plots of stator phase current measurements under healthy and faulty conditions

Moreover, the anomaly detection results in Figures 14 (c-d) for demagnetization and 15 (c-d) for eccentricity further confirm that the CAE model trained with frequency-domain information from phase current measurements struggles to differentiate between healthy and faulty conditions, especially for demagnetization cases. The minimal deviations make it challenging for the CAE model to distinguish faulty samples from healthy ones, resulting in similar anomaly scores.

Time-domain signals, lasting 0.2 seconds, are also presented in Figure 3 for both healthy and various fault conditions. Despite their subtlety, fault-related deviations are apparent in the time-domain data, demonstrating that the CAE model trained with time-domain phase current measurements more effectively detects anomalies, as shown in Figures 11 (c-d) for demagnetization and 12 (c-d) for eccentricity.

[Figure]

Figure 3: Time waveforms of stator phase current measurements under healthy and faulty conditions

In Section 4.2, we assess anomaly detection models based on two criteria: fault detection accuracy and sensitivity to fault severity. Although some measurements provide satisfactory detection results, they lack sufficient sensitivity to variations in fault severity. For instance, while the model trained with electromagnetic torque measurements detects faults with high anomaly scores, as evidenced in Figures 11 and 12 (b), it does not adequately differentiate between levels of fault severity. Consequently, the CAE model trained with time-domain three-phase current measurements is identified as the most effective model in this section, combining high detection accuracy with adequate sensitivity to fault severities.

**Comment#5:**

Does the accuracy of the fault detection is influenced by the sampling frequency of the signal?

**Answer#5:**

Thank you for your comment. As mentioned in response to Comment #4, signal resolution significantly influences the visibility of fault-related deviations in the frequency spectrum of measurements. Two key factors affecting this resolution are the sampling frequency, which determines the Nyquist frequency and the capture of high-frequency information, and the signal length, which influences the resolution of frequency components.

To illustrate the impact of sampling frequency on signal resolution, Figures 4 (a-d) presents PSD plots of the electromagnetic torque signal under both healthy and static eccentricity fault conditions across four different sampling frequencies. These plots demonstrate that adjustments in sampling frequency alter the frequency content visible in the PSD spectrum in accordance with the corresponding Nyquist frequency. However, the resolution of existing frequency information remains constant. A lower sampling frequency may omit high-frequency fault-related variations, potentially impacting the performance of the machine learning model in fault detection. All results presented in this paper were generated using signals simulated at a sampling frequency of *Fs=21.132 kHz*.

Additionally, the effect of signal duration on frequency resolution is explored in Figures 5 (a-d). These figures display PSD spectrums of the electromagnetic torque signal for healthy and static eccentricity fault conditions at a fixed sampling frequency of *Fs=21.132 kHz*, but with varying signal lengths. As shown, shorter signals result in decreased frequency spectrum resolution, making fault-related rises in sideband harmonics less visible.

A comprehensive investigation into the effects of sampling frequency and signal length on the fault detection performance of machine learning models is planned for future work.

[Figure]

Figure 4: Power spectral density plots of electromagnetic torque measurements under healthy and static eccentricity fault conditions with different sampling frequencies

[Figure]

Figure 5: Power spectral density plots of electromagnetic torque measurements under healthy and static eccentricity fault conditions with different signal durations

**Responses to Reviewer #2 comments (RC2):**

**Comment#1:**

General note on use of current for anomaly detection/defect identification - generators are normally analyzed using both voltage and current with voltage being the primary, especially in PMSG. Also, general note, ESA is presently applied in DFIG, induction and PMSG in wind. For the voltage analysis of generators reference Howard Haynes, et.al, mcsa and esa work out of Oak Ridge National Labs (circa 1988-1992) and Penrose post 2003 (IEEE papers) with emphasis post 2010.

**Answer#1:**

We appreciate your insightful remarks and thorough evaluation of our article. Thank you also for providing valuable information regarding the current technologies established in the industry, including applications of current and voltage for condition monitoring and fault detection of electrical machines, along with pertinent references.

Our study indeed recognizes the need for a comprehensive approach to fault detection. Consequently, we have expanded our focus beyond merely current and voltage measurements to include a variety of other electrical and electromagnetic signals such as induced shaft voltage, electromagnetic torque, and airgap/stray magnetic flux sensors. Each of these measurements was carefully selected for its unique potential to contribute to the detection of specific fault types in large MW-scale PMSGs—an area we identified as having substantial room for improvement in diagnostic capabilities, especially for early fault detection.

The motivation behind employing a diverse array of measurements was to evaluate their effectiveness in enhancing the sensitivity and accuracy of fault detection systems, particularly when integrated with advanced machine learning/deep learning techniques, such as the convolutional autoencoder technique utilized in this study. This approach is specifically designed to overcome the limitations associated with the traditional low-resolution data typically found in SCADA systems, which often struggle to capture early-stage fault signatures effectively.

We are grateful again for your valuable suggestions and references, which will certainly guide the future extensions of this work. Integrating additional insights from established ESA and MCSA methodologies could indeed enhance our understanding and lead to the development of more robust diagnostic tools. We look forward to exploring these avenues to better align our research with industry standards and the latest technological advancements in generator condition monitoring.

**Comment#2:**

(53) - what is the FMAX and resolution for the current portion of the study? How are you analyzing static, dynamic and mixed eccentricity which is the function of the number of magnets times the RPM with sidebands for frequency based?

**Answer#2:**

In our simulations, the sampling frequency for current signals and other measurements was set at 21.132 kHz, resulting in a maximum frequency (FMAX) of 10.566 kHz. We believe that the choice of sampling frequency significantly influences the resolution of the signals, which is crucial for accurately capturing the frequency information pertinent to fault detection. This was highlighted in response to Comment#5 from the first reviewer. We conducted an initial analysis on the impact of sampling frequency on the PSD resolution and demonstrated that a lower sampling frequency could potentially omit critical frequency information present in the higher frequency ranges, thus impacting the accuracy of fault detection by the machine learning model. All analyses in this study were conducted with the mentioned fixed sampling frequency. However, we recognize the importance of exploring how varying signal

resolutions affect fault detection capabilities and plan to undertake a comprehensive investigation on this matter in future studies.

Regarding the analysis of static, dynamic, and mixed eccentricity, harmonic analysis has been extensively employed in the literature to investigate such faults. Some examples include studies on fault detection in PM machines: https://doi.org/10.1109/TIE.2009.2029577 and https://doi.org/10.1109/ICEMS.2013.6713349. However, it's important to note that the primary focus of this study was not to replicate conventional techniques such as harmonic analysis. Instead, our objective is to develop an end-to-end framework that utilizes the capabilities of machine learning to automatically capture relevant features and fault-related deviations in both time-domain and frequency-domain data. This is particularly crucial for detecting early-stage failures, which may not be easily detectable or quantifiable using traditional methods.

Additionally, it should be noted that in this study, only static and dynamic eccentricities were simulated in the generator model, and mixed eccentricity states were not included in the analyses. In this regard, in response to Comment#3 from the first reviewer, we have detailed how static and dynamic eccentricities have been simulated in our generator model.

**Comment#3:**

(57) - SCADA has been used to detect conditions, electrical and mechanical, in variable speed induction and synchronous in pharma machines (motors, servos) as infrequently as once per hour (re: IEEE paper 'Machine Learning for Electric Machine Prognostics and Remaining Useful Life with Basic Motor Data). However, CAE (or any NN methodology) was unable to perform well as the work is time-series anomaly detection.

**Answer#3:**

We acknowledge the successful applications of SCADA data in fault detection across various types of electrical machines, as exemplified by the study you referenced. However, the limitations of SCADA data should also be considered, particularly in detecting early-stage faults, where anomalies may present very subtle signatures. The typical SCADA resolution of once per hour or minute may not be sufficiently sensitive to detect such early signs, both electrical and mechanical. For this purpose, the primary motivation of our current study is to explore the potential of higher-resolution measurements combined with machine learning models to enhance the sensitivity and accuracy of fault detection, especially in the early-developing stages of failures.

Furthermore, we investigated different severities of fault cases and assessed the capabilities of various measurements, focusing on two key factors: fault detection accuracy and sensitivity to fault severity. Our findings indicate that while most measurements successfully detected anomalies/faults for both demagnetization and eccentricity, they often lacked sufficient sensitivity to the gradations of fault severity—a critical factor in early-stage fault management. As mentioned earlier, our future studies will focus on investigating different signal resolutions, including those typical of SCADA systems, to determine the minimum required resolution for effective fault detection.

**Comment#4:**

(66) - MCSA/ESA rules-based systems rely upon spectral (frequency) analysis using >10kHz sampling rate per channel with a high enough FMAX (Nyquist is 1/2 sampling, so a minimum 5kHz) to observe static and dynamic eccentricity and stator defects. Stator defects are the number of slots or coils times the running speed with line frequency sidebands and stator electrical defects are found in two types of signals, the stator frequency would have sidebands of running speed and the other would be a 'winding short' signature (re IEEE papers: On-Line Diagnosis of Stator Shorted Turns in Mains and Inverter-Fed Low Voltage Induction Motors', and 'Evaluation of Stator and Rotor Interturn Stress with Electrical Signature Analysis in Variable Frequency Drive and Wind Generator Applications')

(~66) - continued: Some ML/AI system attempts at ESA/MCSA harmonic analysis have been attempted and found to be inaccurate. Rules-based systems have been utilized in the wind industry since 2003 including PMSG systems. While many of the signatures in PMSG will appear to be harmonic in nature, they are actually speed and component related and result in an additive signature onto the harmonic.

**Answer#4:**

Thank you for highlighting the critical aspects of MCSA/ESA rules-based systems and for providing valuable references. We greatly appreciate the information, which enhances our understanding of established methodologies in the field, invaluable for diagnosing various types of electrical machine faults.

We again acknowledge and are aware of the widely used techniques in the condition monitoring of electrical machines, such as frequency harmonic analysis, which we have referenced in the introduction section of the paper and have included some of your provided references as well. However, as previously mentioned, our study aimed to explore the potential of ML/AI to automatically extract meaningful and fault-related features from various measurements. By employing ML models, we sought to determine whether these intelligent methods could offer a complementary approach to traditional techniques, particularly in enhancing the detection capabilities for early-stage faults and subtle anomalies in signal data.

We are also aware of the challenges and limitations regarding the application of ML/AI techniques in this area, such as the requirement for a large amount of data to ensure robustness in terms of accuracy and generalizability. Additionally, there are challenges in implementing these models in real-time scenarios. Thus, there is substantial room for research exploring techniques to optimize ML models for seamless integration into operational industrial systems.

**Comment#5:**

General Note: I noted this was all at fixed speed and frequency. As the data is taken directly from the generator, how are wind speed variations including wind gusts managed?

**Answer#5:**

Yes, all analyses in our study were conducted at a fixed rated wind speed, without considering wind speed variations. As detailed in Section 2 of the manuscript, the generator model is a high-speed PMSG, designed according to the specifications of the NREL 5-MW reference offshore wind turbine drivetrain (see Table 1 in the manuscript). Measurements in this model were simulated at a constant speed of 1173.7 RPM, equivalent to a wind speed of 12.1 RPM with a gearbox ratio of 1:97. This fixed-speed simulation was employed to isolate and accurately measure the impact of various faults on the electrical and electromagnetic signatures of the generator. Managing wind speed variations, including gusts, remains crucial for practical applications, and exploring this aspect is planned for future studies.

**Comment#6:**

Anomaly Detection vs Time Series: with variable loads and speeds, how would severity or RUL be determined? I also assume no classification is being performed outside of static/dynamic, which would include stator/rotor defects?

**Answer#6:**

In this study, anomaly detection is primarily aimed at identifying deviations from normal operational signatures at a fixed speed, focusing on the results of demagnetization and both static and dynamic eccentricity faults. We currently concentrate on distinguishing between healthy conditions and these specific fault types, without addressing other defects. Additionally, our methodology does not cover RUL estimation under either fixed or variable operating

conditions, which is beyond the scope of this paper. However, a literature review suggests that RUL can be estimated under variable conditions using approaches such as physics-based modeling, data-driven algorithms, and hybrid methods that combine both (https://doi.org/10.1109/SPECon61254.2024.10537428, https://doi.org/10.1109/MIE.2020.3016138). These methods adapt to changes in operational profiles by incorporating dynamic data inputs and advanced feature engineering to enhance prediction accuracy and robustness.

**Comment#7:**

(330) - Wouldn't demag of the magnets show in dynamic eccentricity as well? Static and dynamic eccentricity can be magnetic as well as mechanical, especially dynamic eccentricity. I'm not sure if this is a missed consideration or a condition of the model. However, this is generally understood in application.

**Answer#7:**

Thank you for your thoughtful comment on this matter. We value your expertise and recognize that in practical applications, demagnetization can indeed influence the manifestation of dynamic eccentricity, given that both magnetic and mechanical factors contribute to eccentricity, especially dynamic eccentricity.

However, in our model, the faults were specifically designed such that demagnetization does not induce dynamic eccentricity. This approach was chosen to isolate the effects of each fault type on the generator's performance, facilitating a clearer analysis and understanding. As described in response to Comment#3 from the first reviewer, eccentricity cases are simulated in the 2D (xy-plane) to model the non-uniform airgap. For static eccentricity, the stator axis is shifted while the rotor axis remains fixed at the origin (0,0), resulting in a non-uniform airgap where the length distribution does not vary with rotation. In contrast, for dynamic eccentricity, the rotor axis moves during rotation while the stator axis remains fixed, creating a non-uniform airgap with a length distribution that varies during rotation. The degree of eccentricity is quantified by the offset value of the shifted axis relative to the airgap length in a healthy condition.

**Comment#8:**

(336) - How are ground conditions and changes in load/speed, EMI, radio signals, etc. managed with the external flux probes (vs air gap)? These conditions have been an issue in attempted applications since first introduced by companies like CSI (then Emerson) in the 1990s. This includes the sympathetic radiation from the generator from such signals and the shielding generated by the frame.

**Answer#8:**

Thank you for your concern regarding the management of various environmental and operational factors affecting external flux probes. We understand the significance of these issues in practical applications, as historically noted in the field.

In this study, however, the focus was specifically on analyzing the measurements of the generator without extending to external environmental influences or interference. As such, these external factors were not modeled in the current generator model. These aspects remain outside the scope of this paper but represent important directions for future research to ensure the robustness of diagnostic techniques in real-world scenarios.

**Comment#9:**

(360-375) - yes, current still detects these issues, but at a much later stage (reflection of current). Normally ESA, instead of MCSA, would be used to evaluate both voltage and current of the generator. As previously noted, current is the load (downstream) voltage is the source (upstream).

**Answer#9:**

You are correct in noting that current signals, which reflect downstream effects, often detect issues at a later stage compared to the upstream source signals like voltage. Also, regarding the comparison of current signal with magnetic flux measurements, as highlighted in our results, although current signals effectively distinguished between different fault cases, the use of stray flux measurements provided more nuanced insights, particularly in distinguishing between eccentricity and demagnetization faults.

Finally, thank you and the reviewers again for the instructive comments. We greatly appreciate reviewers' thorough and insightful feedback throughout this review process, which has been invaluable in refining our study and guiding our future research directions.

Yours sincerely,

Ali Dibaj
Ph.D. Candidate
Department of Marine Technology (IMT)
Norwegian University of Science and Technology (NTNU)
Trondheim, Norway